

# Establishment and validation of a risk prediction model for adverse drug reactions in patients with coronary heart disease after taking statins: a retrospective study

Lixiang Zhang, Jiaoyu Cao and Xiaojuan Zhou

Department of Cardiology, The First Affiliated Hospital of USTC, Division of Life Science and Medicine, University of Science and Technology of China, Hefei, Anhui Province, China

Corresponding author
Xiaojuan Zhou, 3268584430@qq.com

## ABSTRACT

**Objective**. This study aims to develop and validate a nomogram-based predictive model for estimating the risk of adverse drug reactions (ADR) to statins in patients with coronary heart disease (CHD).

**Methods**. A retrospective cohort study was conducted using clinical data from 351 patients with CHD who received statin therapy in the cardiology department of a tertiary hospital in Anhui Province, China, between February 2021 and January 2022. The dataset was randomly divided into a development cohort ($n = 283$) and a validation cohort ($n = 68$) in an 8:2 ratio. Logistic regression analysis was applied in the development cohort to identify independent risk factors for statin-induced ADR. A nomogram was subsequently constructed in R based on the selected predictors, and its clinical utility, discriminative performance, and calibration were evaluated.

**Results**. The overall incidence of statin-associated ADR among the 351 subjects was 24.22%, classified into three categories according to the affected system: musculoskeletal toxicity, hepatic/renal dysfunction, and gastrointestinal reactions. Univariate and multivariate logistic regression analyses in the development cohort identified the following as significant independent risk factors ($P < 0.05$): age $\geq 60$ years, body mass index $\geq 23$ kg/m$^2$, disease duration $\geq 5$ years, presence of $\geq 3$ comorbid conditions, dyslipidemia, history of cerebral infarction, high-dose statin use, and concomitant use of multiple medications. A nomogram model was constructed based on these predictors. The model demonstrated strong discriminative performance, with an area under the receiver operating characteristic (ROC) curve of 0.808 (95% CI [0.751–0.865]) in the development cohort and 0.852 (95% CI [0.752–0.951]) in the validation cohort.

**Conclusion**. A nomogram-based risk prediction model was successfully developed to estimate the probability of statin-induced ADR in patients with CHD, based on a set of statistically significant clinical risk factors. The model exhibited favorable predictive accuracy and discrimination. It offers a practical tool for clinicians to identify high-risk individuals and implement early preventive or interventional strategies accordingly.

## INTRODUCTION

The incidence of coronary heart disease (CHD) has markedly increased, driven by rapid economic development and shifts in dietary habits, resulting in a progressively younger age of onset. CHD has become a major contributor to global mortality, ranking as the leading cause of death among males aged 45 and older (*Dinicolantonio, Lucan & O'Keefe, 2016*; *Zhang et al., 2023*). Clinical manifestations include chest tightness, dyspnea, and other related symptoms. CHD is associated with numerous complications such as angina, arrhythmia, heart failure, thromboembolism, and myocardial infarction (*Zhang, 2023*; *Liu, Chang & Wang, 2023*). In severe cases, it can lead to cardiac arrest, exerting a profound impact on patients' overall health, functional status, and quality of life (*Ference et al., 2019*).

Currently, pharmacological treatment of CHD primarily relies on lipid-lowering agents, with statins being the first-line therapy for dyslipidemia (*Guo et al., 2020*). Statins exert their therapeutic effect by competitively inhibiting 3-hydroxy-3-methylglutaryl coenzyme A reductase, thereby suppressing the synthesis of endogenous mevalonate and cholesterol, ultimately achieving lipid modulation (*Mei, 2023*; *Karaźniewicz-Łada et al., 2018*). Evidence suggests that statins are widely utilized in clinical practice, effectively lowering blood lipid levels by reducing peripheral lipoproteins and mitigating vascular stenosis in patients with CHD (*Guan et al., 2019*).

However, evidence indicates that patients with CHD may experience varying degrees of adverse drug reactions (ADR) following statin therapy, with myalgia, elevated hepatic enzymes, and renal impairment being among the most frequently reported effects (*Liu et al., 2019*). Previous studies have documented the prevalence and nature of statin-associated adverse drug reactions (ADRs). A United Arab Emirates (UAE)-based retrospective cohort study found a 40.7% incidence of ADRs among statin users, with myopathy and elevated creatine kinase (CK) levels occurring in 15% of cases, underscoring muscular toxicity as a key concern (*Shehab et al., 2020*; *Alzueta et al., 2021*). Additionally, a large-scale study reported that statin use was associated with a higher risk of gastrointestinal bleeding, particularly in the first year of therapy (adjusted risk ratio: 1.19), along with increased hospitalization rates for this complication (adjusted risk ratio: 1.38) (*Martinez, Freeman & Moga, 2019*).

Given the substantial incidence and clinical implications of ADR associated with statin therapy, there is an urgent need to identify predisposing risk factors and implement targeted strategies to mitigate these outcomes. A comprehensive understanding of the determinants contributing to ADR in patients with CHD enables the formulation of individualized interventions to enhance treatment safety and adherence. Accordingly, early identification of high-risk subpopulations is essential for effective risk stratification and the design of preventive measures. This study analyzed the risk factors for ADR in statin-treated patients with CHD and developed a personalized risk prediction model to guide clinical decision-making and reduce the incidence of ADR in this population.

## MATERIALS AND METHODS

### General information

This retrospective cohort study included 351 patients diagnosed with CHD who were admitted to the Department of Cardiology at a tertiary hospital in Anhui Province, China, between February 2021 and January 2022, and who received statin therapy during hospitalization. Inclusion criteria were as follows: (1) Diagnosis of CHD confirmed by coronary angiography, in accordance with established diagnostic criteria (*Shen, Zhang & Shen, 2019*). CHD diagnosis is based on a comprehensive evaluation of clinical symptoms, physical examination findings, electrocardiogram (ECG) results, and imaging assessments such as coronary angiography or coronary computed tomographic angiography (CTA). Clinically, typical angina pectoris is characterized by a sensation of pressure, constriction, or tightness localized retrosternally or in the precordial region, often triggered by physical exertion or emotional stress. The discomfort may radiate to the left shoulder or the medial aspect of the left upper arm, typically lasting several minutes and alleviated by rest or sublingual nitroglycerin administration. Physical examination may reveal systolic murmurs at the cardiac apex or along the left sternal border (third or fourth intercostal space), as well as tachycardia and hypertension during anginal episodes. ECG findings during an episode typically reveal ST-segment depression and T-wave inversion, indicative of myocardial ischemia. In contrast, resting ECGs may appear normal or exhibit residual signs of prior ischemic injury, such as persistent ST-T segment abnormalities. Exercise stress testing is considered positive if typical anginal symptoms are elicited during exertion or if horizontal or downsloping ST-segment depression of $\geq 0.1$ mV is observed during or following exercise. Coronary angiography remains the gold standard for diagnosing CHD, offering precise visualization of the location, severity, and extent of coronary artery stenosis. A CHD diagnosis is established when major coronary arteries—such as the left main, left anterior descending, circumflex, or right coronary artery—exhibit $\geq 50\%$ luminal narrowing. Coronary computed tomography angiography (CTA) also serves as a valuable diagnostic tool, demonstrating high sensitivity and specificity for detecting coronary stenosis and functioning as a noninvasive complement to invasive angiography; (2) absence of hypersensitivity to statins; (3) provision of informed consent by the patient and their family for the treatment protocol and associated precautions; (4) complete clinical data with no missing information; (5) good adherence to the prescribed treatment regimen. Exclusion criteria were as follows: (1) Presence of significant cognitive impairment; (2) diagnosis of malignant neoplasms; (3) presence of organic lesions in critical organs (*e.g.*, liver, kidney, lung); (4) diagnosis of hematologic or autoimmune disorders; (5) incomplete clinical data. The flowchart of participant selection is detailed in Fig. 1. Ethical approval for this study was obtained from the Medical Ethics Committee of the First Affiliated Hospital of the University of Science and Technology of China (Approval No.: 2023-RE-216). Given the retrospective nature of the study, the requirement for informed consent was waived by the committee.

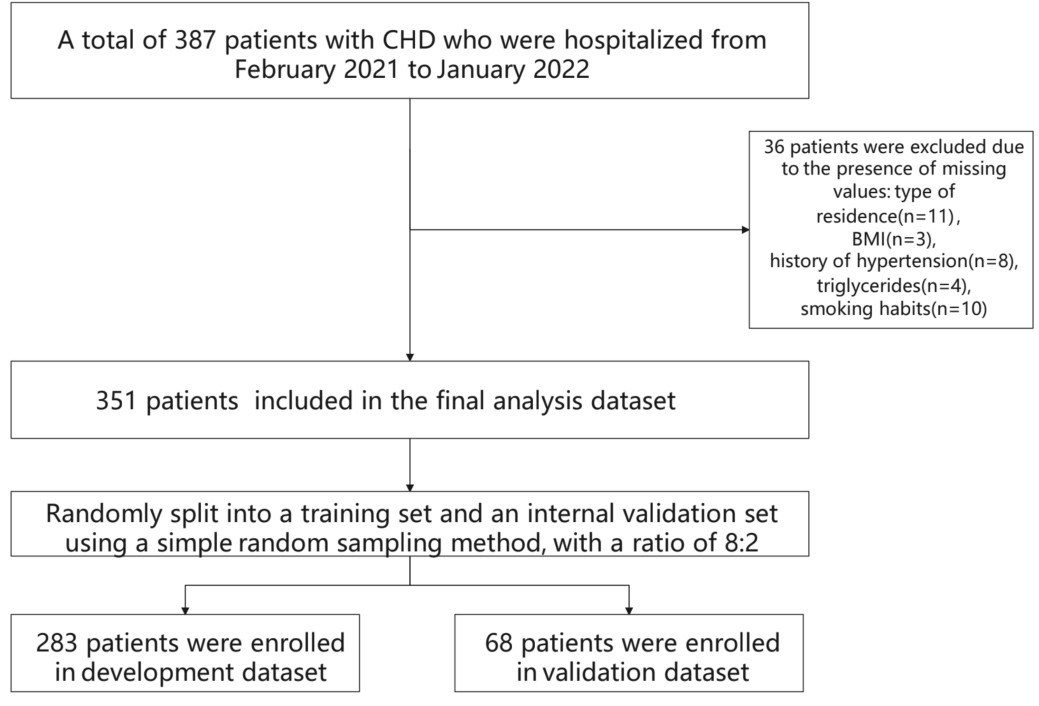

**Figure 1 Flowchart of participant selection.** A total of 387 hospitalized patients with coronary heart disease (CHD), admitted between February 2021 and January 2022, were initially screened. After excluding 36 patients due to missing data—type of residence ($n = 11$), BMI ($n = 3$), hypertension history ($n = 8$), triglyceride levels ($n = 4$), and smoking status ($n = 10$)—351 patients remained eligible for inclusion in the final analysis. These individuals were subsequently allocated into a development cohort ($n = 283$) and a validation cohort ($n = 68$) using simple random sampling at an 8:2 ratio.

## Methods

Clinical data were retrospectively collected and included a comprehensive set of demographic and clinical variables: sex, age, education level, body mass index (BMI), type of residence, medical insurance/payment method, history of hypertension, presence of dyslipidemia (defined as abnormal lipid profiles, including elevated total cholesterol, low-density lipoprotein cholesterol, triglycerides, or decreased high-density lipoprotein cholesterol), smoking status (classified as current, former, or never smokers), alcohol consumption (assessed by frequency and quantity per week), duration of disease, comorbidity profile (including chronic conditions such as hypertension, diabetes, and cardiovascular diseases), history of diabetes, history of cerebral infarction, polypharmacy (defined as concurrent use of multiple medications, quantified by the number and categories of drugs), and high-dose drug exposure (defined as usage exceeding standard recommended therapeutic dosages). A total of 16 variables were collected for each subject.

## Statistical methods

Data analysis was performed using SPSS version 20.0 and R software (version 3.6.1; *R Core Team, 2019*). Categorical variables were compared between groups using the Pearson Chi-square test. Logistic regression analysis was employed to identify independent risk

factors associated with ADR occurrence. Significant predictors identified in the multivariate model ($P < 0.05$) were used to construct a nomogram using the "rms" package in R. The nomogram development process involved: (1) Selecting statistically significant variables from the multivariate logistic regression model ($P < 0.05$); (2) assigning point values to each predictor according to the magnitude of its regression coefficient; (3) constructing a graphical, scaled representation of the model to visually depict the contribution of each factor; and (4) validating the model internally or externally to assess its predictive performance and generalizability. The nomogram serves as a visual, user-friendly tool that integrates multiple risk indicators into a single predictive framework, enabling individualized risk estimation and aiding in early clinical decision-making (*Lu & Li, 2021*; *Lee et al., 2019*; *Ji et al., 2023*). Model performance was evaluated in three dimensions: (1) Discrimination, measured by the area under the receiver operating characteristic (ROC) curve (AUC); (2) calibration, assessed *via* the Hosmer-Lemeshow goodness-of-fit test and calibration curve; (3) clinical utility, examined through decision curve analysis (DCA). All statistical tests were two-tailed, and a *P*-value <0.05 was considered indicative of statistical significance.

# RESULTS

## Characteristics of ADR after using statins in patients with CHD

A total of 351 patients were included in the study. The overall incidence of statin-associated ADR among the 351 subjects was 24.22% (85/351). ADR were further categorized by affected system into three types: musculoskeletal toxicity ($n = 19$, 22.4%), hepatic and renal dysfunction ($n = 26$, 30.6%), and gastrointestinal reactions ($n = 40$, 47.0%). Among the 85 patients who developed ADR, three types of statins were administered: simvastatin ($n = 20$, 23.5%), atorvastatin ($n = 28$, 33.0%), and rosuvastatin ($n = 37$, 43.5%).

## Comparative analysis of clinical data between the development cohort and the validation cohort

The cohort comprised 139 female patients (39.60%) and 212 male patients (60.40%). A total of 238 patients (67.81%) were aged 60 years or older. Additional clinical characteristics of the study population are summarized in Table 1. There were no statistically significant differences between the development and validation cohorts across baseline characteristics ($P > 0.05$), indicating a high degree of homogeneity between the two datasets (Table 1).

## Results of logistic regression analysis in development cohort

Univariate and multivariate logistic regression analyses conducted in the development cohort identified eight variables as independent risk factors for statin-associated ADR in patients with CHD: age $\geq 60$ years, BMI $\geq 23$ kg/m$^2$, disease duration $\geq 5$ years, number of comorbid conditions $\geq 3$, dyslipidemia, history of cerebral infarction, high-dose statin use, and polypharmacy ($P < 0.05$), as detailed in Table 2.

**Table 1 Comparison of clinical data between the development group and the validation group.** A detailed comparison of various clinical characteristics between the development group and the validation group in the study is presented. The variables include high-dose medication use, combined medication, history of cerebral infarction, diabetes, basic disease type, disease course duration, alcohol consumption, smoking status, dyslipidemia, hypertension, payment method for medical expenses, type of residence, body mass index (BMI), education level, age, gender, and the occurrence of adverse drug reactions (ADR). The total number of participants is 351, with 283 in the development group and 68 in the validation group. For each variable, the number and percentage of participants in each category are provided for both groups, along with the chi-square ($\chi^2$) statistic and corresponding $p$-value to assess the statistical significance of differences between the groups. The results show that most variables are well-balanced between the two groups, with no significant differences observed for the majority of the clinical characteristics.

| Variables | Classification items | Total ($n = 351$) | Development group ($n = 283$) | Validation group ($n = 68$) | $\chi^2$ | $P$ |
|---|---|---|---|---|---|---|
| High dose medication, n (%) | No | 260 (74.07) | 208 (73.50) | 52 (76.47) | 0.252 | 0.616 |
| | Yes | 91 (25.93) | 75 (26.50) | 16 (23.53) | | |
| Combined medication, n (%) | No | 198 (56.41) | 159 (56.18) | 39 (57.35) | 0.030 | 0.861 |
| | Yes | 153 (43.59) | 124 (43.82) | 29 (42.65) | | |
| History of cerebral infarction, n (%) | No | 271 (77.21) | 223 (78.80) | 48 (70.59) | 2.100 | 0.147 |
| | Yes | 80 (22.79) | 60 (21.20) | 20 (29.41) | | |
| History of diabetes, n (%) | No | 149 (42.45) | 119 (42.05) | 30 (44.12) | 0.096 | 0.757 |
| | Yes | 202 (57.55) | 164 (57.95) | 38 (55.88) | | |
| Basic disease type, n (%) | ≥3 | 240 (68.38) | 193 (68.20) | 47 (69.12) | 0.021 | 0.884 |
| | <3 | 111 (31.62) | 90 (31.80) | 21 (30.88) | | |
| Course of disease, n (%) | ≥5 years | 231 (65.81) | 188 (66.43) | 43 (63.24) | 0.249 | 0.618 |
| | <5 years | 120 (34.19) | 95 (33.57) | 25 (36.76) | | |
| Alcohol consumption, n (%) | No | 243 (69.23) | 195 (68.90) | 48 (70.59) | 0.073 | 0.787 |
| | Yes | 108 (30.77) | 88 (31.10) | 20 (29.41) | | |
| Smoking, n (%) | No | 240 (68.38) | 194 (68.55) | 46 (67.65) | 0.021 | 0.886 |
| | Yes | 111 (31.62) | 89 (31.45) | 22 (32.35) | | |
| Dyslipidemia, n (%) | No | 123 (35.04) | 106 (37.46) | 17 (25.00) | 3.737 | 0.053 |
| | Yes | 228 (64.96) | 177 (62.54) | 51 (75.00) | | |
| History of hypertension, n (%) | No | 246 (70.09) | 198 (69.96) | 48 (70.59) | 0.010 | 0.920 |
| | Yes | 105 (29.91) | 85 (30.04) | 20 (29.41) | | |
| Payment method for medical expenses, n (%) | Medical insurance | 294 (83.76) | 233 (82.33) | 61 (89.71) | 2.192 | 0.139 |
| | Self funded | 57 (16.24) | 50 (17.67) | 7 (10.29) | | |
| Type of residence, n (%) | Countryside | 244 (69.52) | 193 (68.20) | 51 (75.00) | 1.197 | 0.274 |
| | City | 107 (30.48) | 90 (31.80) | 17 (25.00) | | |
| Body mass index, n (%) | <23 kg/m$^2$ | 137 (39.03) | 110 (38.87) | 27 (39.71) | 0.016 | 0.899 |
| | ≥23 kg/m$^2$ | 214 (60.97) | 173 (61.13) | 41 (60.29) | | |
| Education level, n (%) | Junior high school and below | 226 (64.39) | 185 (65.37) | 41 (60.29) | 0.690 | 0.708 |
| | High school and technical secondary school | 76 (21.65) | 59 (20.85) | 17 (25.00) | | |
| | College degree or above | 49 (13.96) | 39 (13.78) | 10 (14.71) | | |
| Age, n (%) | ≥60 years old | 238 (67.81) | 193 (68.20) | 45 (66.18) | 0.103 | 0.749 |
| | <60 years old | 113 (32.19) | 90 (31.80) | 23 (33.82) | | |
| Gender, n (%) | Female | 139 (39.60) | 114 (40.28) | 25 (36.76) | 0.284 | 0.594 |
| | Male | 212 (60.40) | 169 (59.72) | 43 (63.24) | | |
| ADR, n (%) | Not Occurred | 266 (75.78) | 215 (75.97) | 51 (75.00) | 0.028 | 0.867 |
| | Occurrence | 85 (24.22) | 68 (24.03) | 17 (25.00) | | |

**Table 2 Results of univariate and multivariate logistic regression analysis in development group.** The table presents the results of univariate and multivariate logistic regression analyses conducted to identify risk factors associated with adverse drug reactions (ADR) in the development group. The variables included in the analysis cover demographic characteristics (gender, age, education level), lifestyle factors (smoking, alcohol consumption), clinical history (history of hypertension, diabetes, cerebral infarction), disease-related factors (course of disease, basic disease types), and treatment-related factors (combined medication, high-dose medication). For each variable, the odds ratio (OR) and 95% confidence interval (CI) are provided for both univariate and multivariate analyses, along with the corresponding $p$-values. In the multivariate analysis, variables such as age ($\geq$60 years old), body mass index ($\geq$23 kg/m$^2$), dyslipidemia, course of disease ($\geq$5 years), basic disease types ($\geq$3), history of diabetes, history of cerebral infarction, combined medication, and high-dose medication were found to be significant predictors of ADR, with $p$-values less than 0.05.

| Variables | Univariate logistic regression | | Multivariate logistic regression | |
|---|---|---|---|---|
| | OR (95% CI) | *P*-value | OR (95% CI) | *P*-value |
| Gender | | | | |
| Male | 1.00 (Reference) | | | |
| Female | 1.33 (0.77–2.31) | 0.307 | | |
| Age | | | | |
| <60 years old | 1.00 (Reference) | | 1.00 (Reference) | |
| ≥60 years old | 2.36 (1.21–4.59) | 0.011 | 2.20 (1.02–4.71) | 0.043 |
| Education level | | | | |
| Junior high school and below | 1.00 (Reference) | | | |
| High school and technical secondary school | 0.84 (0.41–1.73) | 0.642 | | |
| College degree or above | 1.65 (0.78–3.49) | 0.189 | | |
| Body mass index | | | | |
| <23 kg/m$^2$ | 1.00 (Reference) | | 1.00 (Reference) | |
| ≥23 kg/m$^2$ | 2.52 (1.36–4.70) | 0.004 | 2.89 (1.40–5.96) | 0.004 |
| Type of residence | | | | |
| Countryside | 1.00 (Reference) | | | |
| City | 0.95 (0.52–1.70) | 0.852 | | |
| Payment method for medical expenses | | | | |
| Self funded | 1.00 (Reference) | | | |
| medical insurance | 0.54 (0.28–1.06) | 0.072 | | |
| History of hypertension | | | | |
| No | 1.00 (Reference) | | | |
| Yes | 0.72 (0.39–1.34) | 0.300 | | |
| Dyslipidemia | | | | |
| No | 1.00 (Reference) | | 1.00 (Reference) | |
| Yes | 2.12 (1.15–3.91) | 0.016 | 2.42 (1.18–4.95) | 0.015 |
| Smoking | | | | |
| No | 1.00 (Reference) | | | |
| Yes | 1.15 (0.65–2.06) | 0.629 | | |
| Alcohol consumption | | | | |
| No | 1.00 (Reference) | | | |
| Yes | 1.08 (0.60–1.94) | 0.797 | | |
| Course of disease | | | | |
| <5 years | 1.00 (Reference) | | 1.00 (Reference) | |
| ≥5 years | 2.09 (1.11–3.96) | 0.023 | 2.43 (1.15–5.14) | 0.020 |

**Table 2** (*continued*)

| Variables | Univariate logistic regression | | Multivariate logistic regression | |
|---|---|---|---|---|
| | OR (95% CI) | *P*-value | OR (95% CI) | *P*-value |
| Basic disease types | | | | |
| <3 | 1.00 (Reference) | | 1.00 (Reference) | |
| ≥3 | 2.11 (1.10–4.04) | 0.025 | 2.20 (1.03–4.70) | 0.041 |
| History of diabetes | | | | |
| No | 1.00 (Reference) | | 1.00 (Reference) | |
| Yes | 1.87 (1.05–3.35) | 0.034 | 1.88 (0.95–3.72) | 0.070 |
| History of cerebral infarction | | | | |
| No | 1.00 (Reference) | | 1.00 (Reference) | |
| Yes | 2.02 (1.08–3.75) | 0.027 | 2.20 (1.04–4.66) | 0.040 |
| Combined medication | | | | |
| No | 1.00 (Reference) | | 1.00 (Reference) | |
| Yes | 3.09 (1.75–5.47) | <0.001 | 3.85 (1.97–7.54) | <0.001 |
| High dose medication | | | | |
| No | 1.00 (Reference) | | 1.00 (Reference) | |
| Yes | 3.56 (1.99–6.36) | <0.001 | 4.47 (2.24–8.91) | <0.001 |

### Establishment of a risk model for predicting the risk of ADR in patients with CHD after taking statins

In the development cohort, a nomogram model was constructed using univariate and multivariate logistic regression analyses to identify eight independent risk factors associated with statin-induced ADR in patients with CHD. The model is illustrated in Fig. 2. The optimal cut-off value for the total nomogram score was determined to be 32.05, yielding a sensitivity of 0.749, specificity of 0.706, positive predictive value of 0.890, and negative predictive value of 0.471.

### Clinical applicability of the nomogram for risk of ADR in patients with CHD after taking statins

The DCA was utilized to evaluate the clinical applicability of the nomogram model. The analysis demonstrated that the nomogram provides the highest net clinical benefit when the threshold probability for predicting ADR in statin-treated patients with CHD falls between 0.03 and 0.83. Within this range, the model consistently outperforms both the "Treat All" and "Treat None" strategies. These findings indicate that the nomogram offers substantial clinical utility in guiding personalized risk-based interventions, as illustrated in Fig. 3.

### Evaluation of discrimination and calibration of the nomogram for risk of ADR in patients with CHD after taking statins

The discriminative performance of the nomogram model was evaluated using the AUC. The model achieved an AUC of 0.808 (95% CI [0.751–0.865]) in the development cohort and 0.852 (95% CI [0.752–0.951]) in the validation cohort, reflecting strong discriminatory ability (Fig. 4). Calibration was assessed using the Hosmer–Lemeshow goodness-of-fit test, which yielded non-significant results in both the development and validation cohorts

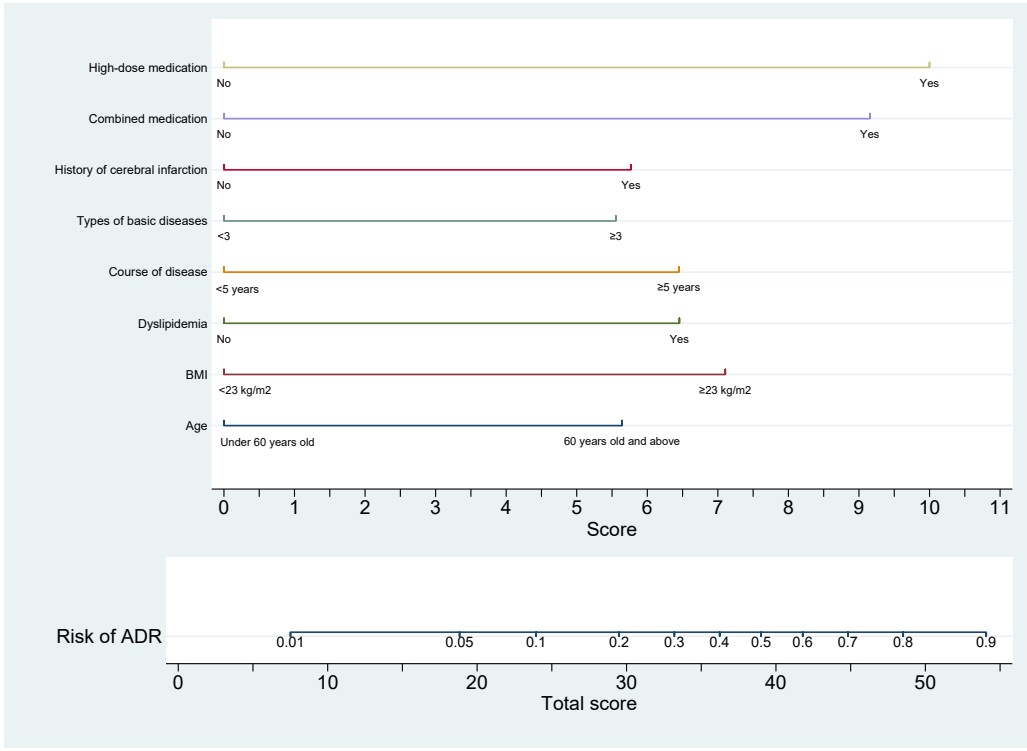

**Figure 2 Nomogram prediction model for statin-induced ADR risk in patients with CHD.** This nomogram provides a quantitative tool for estimating the risk of adverse drug reactions (ADR) in patients with coronary heart disease (CHD) receiving statin therapy. The model integrates eight clinical predictors: age ($<60$ years or $\geq60$ years), BMI ($<23$ kg/m$^2$ or $\geq23$ kg/m$^2$), presence of dyslipidemia (yes/no), duration of disease ($<5$ years or $\geq5$ years), number of comorbid conditions, history of cerebral infarction (yes/no), polypharmacy (yes/no), and high-dose statin use (yes/no). Each variable is assigned a point value (ranging from 0 to 11), and the total score is derived by summing individual scores. The total score is mapped to a corresponding predicted probability of ADR on the upper axis, ranging from 0 to 1.0. This nomogram enables individualized risk assessment, thereby supporting clinicians in tailoring therapeutic strategies and enhancing safety in statin-treated patients with CHD.

($P > 0.05$), indicating no meaningful deviation between predicted and observed ADR probabilities. These findings confirm that the nomogram model is well-calibrated, as shown in Fig. 5.

## DISCUSSION

Statins serve as essential secondary preventive agents in the management of CHD and play a pivotal role in its adjunctive treatment (*Li et al., 2018*). Previous research has reported that approximately 30% of patients with CHD develop ADR following statin therapy (*Li, Lala & Huang, 2016*). In the present study, the observed ADR incidence was 24.22%, modestly lower than previously reported figures. This discrepancy (24.22% *vs.* 30%) may be attributed to several factors. Variations in study populations—including age distribution, comorbid conditions, and genetic predispositions—can significantly influence ADR susceptibility. Additionally, a relatively smaller sample size may reduce the

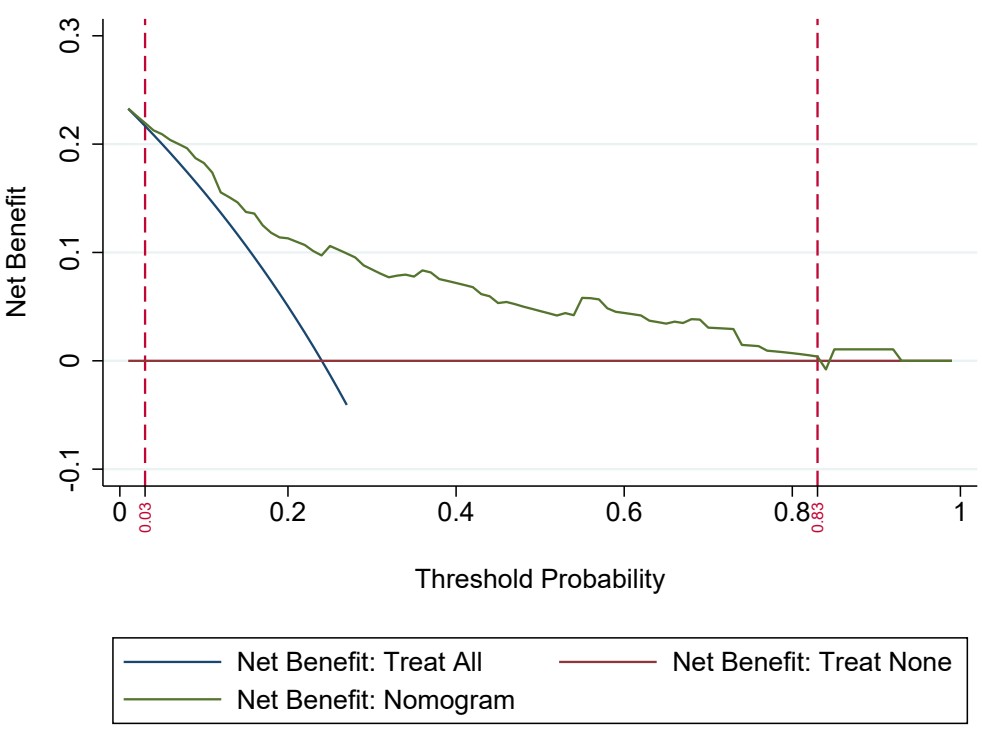

**Figure 3** **Decision curve analysis (DCA) of the nomogram-based prediction model.** The DCA assesses the model's clinical utility by comparing its net benefit to the default strategies of treating all patients *versus* treating none. The *x*-axis denotes the threshold probability (0–1), representing the risk level at which clinical intervention would be considered. The *y*-axis indicates net benefit, reflecting the advantage of applying the nomogram relative to the alternative strategies. The "Net Benefit: Nomogram" curve displays the net benefit across varying thresholds, while the "Treat All" and "Treat None" curves represent the corresponding benefits of universal treatment and no treatment, respectively. Superior net benefit of the nomogram curve across a range of threshold probabilities supports its value in guiding risk-based management of adverse drug reactions (ADR) in patients with coronary heart disease (CHD) receiving statin therapy.

external validity of the findings. Differences in ADR definitions, such as more stringent diagnostic criteria or potential underreporting, could further contribute to the lower observed incidence. Variability in clinical practices, including statin dosing strategies, monitoring protocols, and concomitant medication use, may also affect ADR rates across different settings. Despite the slightly reduced incidence, the data continue to underscore a substantial burden of ADR among patients with CHD receiving statin therapy. Therefore, comprehensive evaluation of the risk factors associated with statin-induced ADR remains crucial for identifying high-risk subgroups and enabling timely, targeted interventions to improve clinical prognosis.

In this study, 85 patients experienced ADR associated with statin therapy. The predominant ADR included gastrointestinal disturbances (40 cases), hepatic and renal dysfunction (26 cases), and statin-induced muscle toxicity (19 cases). According to existing literature, the incidence of abnormal liver function among statin users is

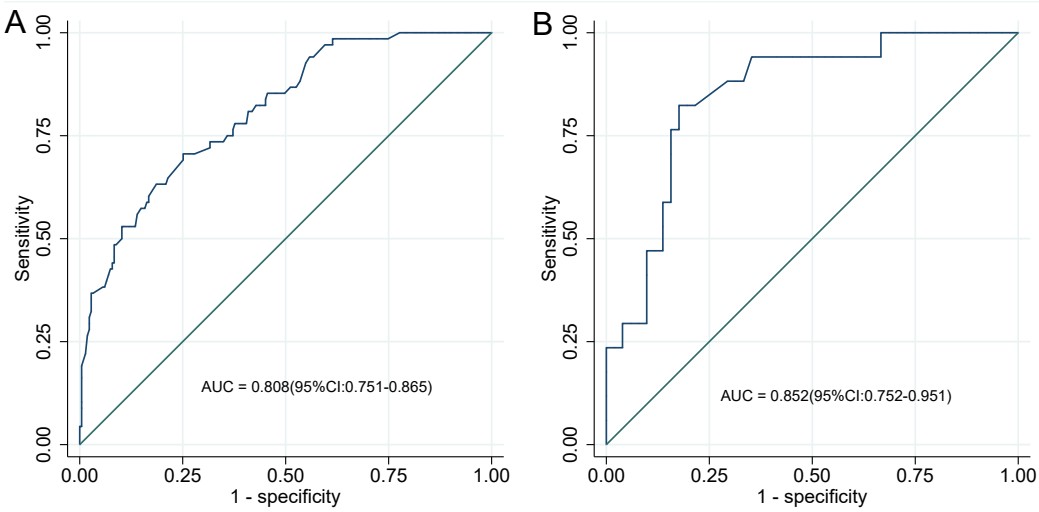

**Figure 4** **ROC curve of the nomogram model.** (A) ROC curve of the nomogram in the development cohort; (B) ROC curve in the validation cohort. ROC curves for the nomogram prediction model are depicted for both development (A) and validation (B) cohorts. The ROC curve illustrates diagnostic performance by plotting sensitivity (true positive rate) against 1-specificity (false positive rate). In the development cohort, the nomogram achieved an AUC of 0.808 (95% CI [0.751–0.865]), while the validation cohort yielded an AUC of 0.852 (95% CI [0.752–0.951]). These AUC values reflect the model's discriminative accuracy in identifying adverse drug reactions (ADR) among patients with coronary heart disease (CHD) receiving statin therapy, with higher AUCs indicating superior predictive capability.

relatively low, generally reported between 0.5% and 2.0% (*Ramkumar, Raghunath & Raghunath, 2016*). Gastrointestinal symptoms, particularly upper gastrointestinal issues such as gastroesophageal reflux, dyspepsia, and esophagitis, are frequently observed in statin-treated patients, with reported incidence rates reaching up to 7% (*Tsui et al., 2023*). Statin-associated muscle toxicity has been reported to occur in 1.5% to 10% of cases (*Auer et al., 2016*). Among the 351 participants in this study, the incidence rates for gastrointestinal reactions, hepatic and renal dysfunction, and muscle toxicity were 11.4%, 7.4%, and 5.4%, respectively. Except for muscle toxicity, these rates exceed those reported in previous studies. Several factors may account for this elevation. First, the study population may have had a higher prevalence of comorbidities or baseline health conditions, predisposing them to increased ADR susceptibility. Second, variations in statin type, dosage, or treatment regimens might have influenced the frequency of adverse outcomes. Additionally, methodological differences, including the sample size and data collection approach, may have contributed to a higher observed incidence. Statin therapy is associated with muscle toxicity, which may manifest as localized or diffuse myalgia, muscle weakness, and elevated serum creatine kinase levels. In severe cases, it can progress to rhabdomyolysis (*Gui et al., 2017*). Rhabdomyolysis results in the release of myoglobin into the bloodstream, imposing significant stress on the kidneys and potentially leading to complications such as renal dysfunction (*Li et al., 2020*). Additionally, evidence suggests that elevated transaminase levels and hepatocellular injury represent the most frequently

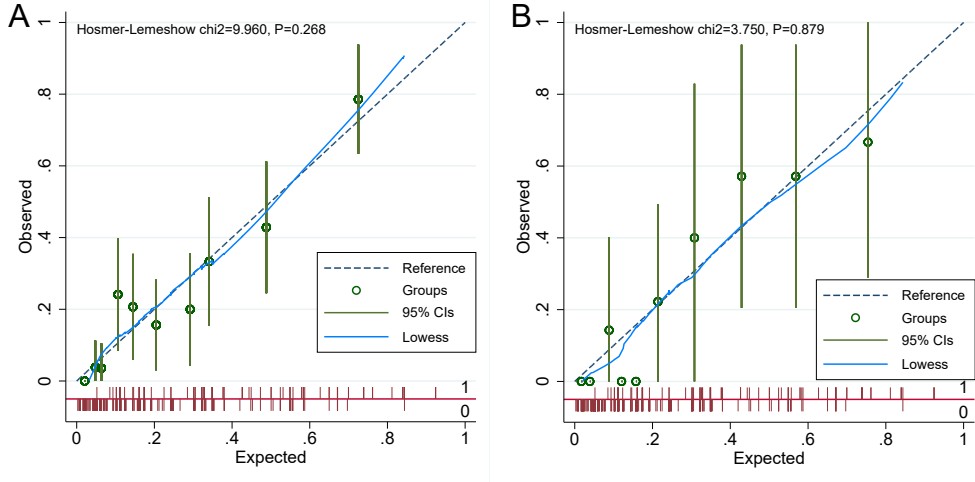

**Figure 5 Calibration curve of the nomogram model.** (A) Calibration curve for the development cohort; (B) calibration curve for the validation cohort. The calibration curves of the nomogram prediction model are presented in both the development (A) and validation (B) cohorts, assessing the agreement between predicted and observed probabilities of adverse drug reactions (ADR) in patients with coronary heart disease (CHD) receiving statin therapy. The *x*-axis represents the predicted probabilities generated by the nomogram, while the *y*-axis denotes the observed incidence of ADR. The 45-degree diagonal line serves as the reference for perfect calibration, where predicted probabilities exactly match actual outcomes. (A) Calibration curve for the development cohort, with a Hosmer-Lemeshow chi-square statistic of 9.960 and a *p*-value of 0.268. (B) Calibration curve for the validation cohort, yielding a chi-square statistic of 3.750 and a *p*-value of 0.879. In both panels, observed *versus* predicted probabilities are plotted alongside 95% CIs and a Lowess smoothing line to illustrate the calibration trend. The close alignment between observed and predicted values indicates the nomogram's predictive accuracy. Non-significant Hosmer-Lemeshow *p*-values (>0.05) in both cohorts suggest good model calibration in estimating the risk of adverse drug reactions (ADR) among patients with coronary heart disease (CHD) undergoing statin therapy.

observed ADR associated with statin use in cardiovascular disease management (*Yang & Bai, 2018*). An observational study on statin-related ADR reported a 9.52% incidence of gastrointestinal symptoms—including dyspepsia, nausea, vomiting, constipation, and abdominal distension—following statin administration (*Chen, 2018*).

This study identified several independent risk factors for ADR following statin therapy in patients with CHD, including age ≥60 years, presence of ≥3 comorbid conditions, history of cerebral infarction, high-dose statin use, polypharmacy, BMI ≥23 kg/m$^2$, disease duration ≥5 years, and dyslipidemia ($P < 0.05$). Advanced age has long been recognized as a significant risk factor for ADR in statin-treated patients with CHD. With aging, the prevalence of comorbidities—such as hypertension, hyperlipidemia, and diabetes—increases markedly due to physiological decline (*Hou et al., 2018*). A higher burden of comorbidities is typically accompanied by impaired hepatic and renal function, reduced drug clearance, and diminished pharmacodynamic response, collectively heightening the risk of ADR (*Holman et al., 2017*).

The elevated risk associated with a history of cerebral infarction and high-dose statin use may reflect the clinical severity of CHD in these patients. High-dose statins are frequently

prescribed in acute or severe cardiovascular conditions, where prior cerebrovascular events often prompt intensified lipid-lowering strategies (*Fu et al., 2020*). However, high-dose statin regimens have been linked to a significantly increased risk of muscle-related toxicity and overall ADR incidence (*Li, 2017*). Notably, dose-escalation studies have demonstrated that the risk of statin-induced myopathy may increase up to fivefold with higher atorvastatin dosages (*Ortega-Alonso et al., 2016*).

Polypharmacy also emerged as a key contributor to ADR risk, primarily due to pharmacokinetic and pharmacodynamic interactions between statins and concomitant medications. Statins are extensively metabolized by hepatic enzymes, particularly cytochrome P450 isoforms. Co-administration with drugs that inhibit or induce these enzymes can disrupt metabolic pathways, thereby increasing systemic drug levels and toxicity. Evidence suggests that over 60% of statin-associated ADR are attributable to such drug interactions (*Liu, 2019*). Agents like atorvastatin and simvastatin may competitively inhibit hepatic transport proteins and metabolic enzymes, potentiating interactions with other substrates (*Pedro-Botet et al., 2016*). Therefore, when statins are prescribed alongside other medications, it is critical to assess the metabolic pathways involved. Failing to do so may elevate the risk of severe complications, including rhabdomyolysis, hepatotoxicity, and other serious ADR (*Li, Tai & Feng, 2023*).

Patients with dyslipidemia, elevated BMI, and prolonged disease duration may encounter specific challenges during statin therapy. Dyslipidemic individuals—particularly those with hyperlipidemia—frequently present with nonalcoholic fatty liver disease (NAFLD), a condition that independently contributes to hepatic dysfunction. In such cases, statin administration may exert an amplified hepatotoxic effect, potentially exacerbating existing liver impairment (*Zheng & Li, 2017*). Elevated BMI has also been shown to correlate with the severity of CHD (*Cui, 2021*), with higher BMI levels associated with greater degrees of coronary artery stenosis (*Cui, 2021*). In severe cases, more intensive lipid-lowering regimens are often employed, thereby increasing the risk of ADR. Furthermore, BMI, disease duration, and age exhibit a positive association with the number of CHD-related comorbidities (*Li, 2021*). As disease duration lengthens, comorbidity burden increases, frequently accompanied by progressive decline in hepatic and renal function, which in turn elevates the likelihood of ADR during statin therapy (*Holman et al., 2017*).

A nomogram is a visual predictive tool derived from multivariate regression analysis, representing multiple clinical variables through a system of calibrated line segments. It facilitates individualized risk estimation by offering an intuitive and continuous method of quantifying the probability of specific clinical outcomes or adverse events (*Zhang, Zou & Dong, 2021*). Nomograms have demonstrated utility in predicting various cardiovascular risks, including new-onset atrial fibrillation during hospitalization for acute coronary syndrome (*Li et al., 2021*), heart failure following percutaneous coronary intervention (PCI) in patients with CHD (*Cao et al., 2022*), and post-PCI bleeding risk in patients with CHD and atrial fibrillation (*Zhao, Ding & Luo, 2020*). However, limited research exists on the application of nomograms for predicting ADR in patients with CHD undergoing statin therapy. This study bridged that gap by constructing a personalized nomogram model based on identified independent risk factors for statin-related ADR in patients with CHD. The

model's performance was rigorously evaluated using the AUC, the Hosmer–Lemeshow goodness-of-fit test, and DCA. The results demonstrated robust discriminative ability, satisfactory calibration, and strong clinical applicability, supporting its potential as a practical tool for individualized ADR risk prediction in this patient population.

The study yields several clinically actionable insights for optimizing statin therapy in patients with CHD: (1) Risk stratification: the nomogram identifies eight independent predictors of statin-associated ADR: age $\geq$ 60 years, BMI $\geq$ 23 kg/m$^2$, disease duration $\geq$ 5 years, dyslipidemia, prior cerebral infarction, high-dose statin use, polypharmacy, and the presence of three or more comorbidities. Patients exhibiting these characteristics warrant heightened surveillance and, when appropriate, consideration of alternative lipid-lowering approaches; (2) Individualized risk estimation: the nomogram offers a quantitative framework for evaluating ADR risk (Fig. 1). A total score $\geq$40 corresponds to a predicted ADR probability exceeding 50%. Integrating this tool into treatment planning facilitates informed risk-benefit assessments, particularly within high-risk subgroups. (3) Monitoring protocols: given the predominance of muscle-related and hepatic/renal toxicities, baseline and serial assessments of CK, alanine aminotransferase (ALT), and serum creatinine are recommended, particularly in high-risk individuals. Early detection of gastrointestinal symptoms through systematic evaluation may also prevent therapy discontinuation due to tolerability concerns; (4) statin selection and dosing: caution is warranted with high-dose regimens and the use of rosuvastatin, which was most frequently associated with ADR in the studied cohort. For at-risk patients, lower doses or substitution with pravastatin or fluvastatin may enhance tolerability profiles; (5) risk communication and shared decision-making: the nomogram's predictive accuracy (AUC 0.808–0.852) supports its use in patient consultations to facilitate discussions on adherence, anticipated ADR, and early symptom reporting. The DCA (Fig. 2) demonstrates clinical utility across a probability threshold of 3–83%, reinforcing its value in routine care; (6) calibration validity: the model's calibration is robust (Hosmer-Lemeshow $P > 0.05$; Fig. 4), supporting reliable risk stratification. Nonetheless, predictions should be interpreted alongside individual factors—such as genetic predispositions—not encompassed by the model; (7) integration into clinical practice: embedding the nomogram into electronic health record (EHR) systems may enable automated risk calculations and support dynamic management adjustments during follow-up. Proactive intervention in high-risk individuals could reduce ADR-related morbidity and enhance long-term adherence to statin therapy. Incorporation of this model promotes precision medicine in lipid management, ensuring therapeutic safety while preserving cardiovascular efficacy.

This study presents several limitations. It employed a retrospective design to identify risk factors associated with statin-induced ADR in patients with CHD, subsequently constructing a predictive model based on these findings. However, the relatively small sample size, restricted number of included variables, and exclusion of patients due to incomplete data may introduce selection bias and limit generalizability. Additionally, both the development and validation cohorts were derived from the same dataset, resulting in a lack of external heterogeneity. Incorporating temporally or geographically distinct datasets for external validation—such as out-of-phase or multicenter cohorts—could enhance

the model's robustness and applicability (*Niu et al., 2023*). Consequently, future research should prioritize prospective, multicenter studies with larger sample sizes and an expanded array of predictive variables to strengthen model validity.

## CONCLUSION

This study identified a 24.22% incidence of statin-associated ADR among patients with CHD, predominantly presenting as myotoxicity, hepatic/renal impairment, and gastrointestinal disturbances. Eight independent risk factors were identified—age $\geq$ 60 years, BMI $\geq$ 23 kg/m$^2$, disease duration $\geq$ 5 years, $\geq$3 comorbidities, dyslipidemia, prior cerebral infarction, high-dose statin therapy, and polypharmacy—and integrated into a nomogram-based prediction model. The model exhibited strong discriminative capability, with AUCs of 0.808 in the development cohort and 0.852 in the validation cohort. This tool supports early identification of high-risk patients and enables risk-informed, individualized interventions to reduce ADR incidence and improve clinical outcomes.

### Funding

This study was funded by the University-level Undergraduate Quality Engineering project of China University of Science and Technology in 2022 (No. 2022xjyxm089). The funders had no role in study design, data collection and analysis, decision to publish, or preparation of the manuscript.

### Grant Disclosures

The following grant information was disclosed by the authors:
China University of Science and Technology: 2022xjyxm089.

### Competing Interests

The authors declare there are no competing interests.

### Author Contributions

- Lixiang Zhang conceived and designed the experiments, analyzed the data, prepared figures and/or tables, authored or reviewed drafts of the article, and approved the final draft.
- Jiaoyu Cao conceived and designed the experiments, performed the experiments, prepared figures and/or tables, authored or reviewed drafts of the article, and approved the final draft.
- Xiaojuan Zhou performed the experiments, analyzed the data, authored or reviewed drafts of the article, and approved the final draft.

### Ethics

The following information was supplied relating to ethical approvals (i.e., approving body and any reference numbers):

The research protocol has received approval from the Medical Ethics Committee at the First Affiliated Hospital of the University of Science and Technology of China (Approval number: 2023-RE-216).

## Data Availability

Raw data is available in the Supplemental Files.

## Supplemental Information

Supplemental information for this article can be found online at http://dx.doi.org/10.7717/peerj.19630#supplemental-information.

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
