# Peer review of "Establishment and validation of a risk prediction model for adverse drug reactions in patients with coronary heart disease after taking statins: a retrospective study"

_PeerJ, doi:10.7717/peerj.19630_

## Round 0.1 · original submission · Major Revisions

The study entitled “Establishment and validation of a risk prediction model for adverse drug reactions in patients with coronary heart disease after taking statins” demonstrated excellent findings using an appropriate methodological approach. However, some important points must be clarified in the manuscript. Your article has great potential for publication on PeerJ, but the reviewers have requested substantial changes to be made, mainly in methodology and discussion sessions.

Reviewer 1 ·

Basic reporting

In the manuscript titled "Establishment and Validation of a Risk Prediction Model for Adverse Drug Reactions in Patients with Coronary Heart Disease After Taking Statins," Lixiang Zhang et al. conducted a retrospective study to identify risk factors for adverse drug reactions (ADRs) and established a nomogram risk prediction model for ADRs. Generally speaking, the story is interesting and well presented. The experimental design is reasonable, the data analysis seems correct, and the results support the claims. Some specific comments are listed as follows:
1. There are slight grammar issues in the manuscript; please correct them. For example, in lines 27-28: "351 individuals diagnosed with coronary heart disease who have been diagnosed with coronary heart disease."
2. In lines 58-60, you may need a reference to support the claim.
3. I would suggest including more background information about the incidence of ADRs in the introduction.
4. I would suggest reorganizing the introduction section into several paragraphs.
5. Detailed figure and table legends are needed for better understanding. Currently, I do not see any legends.
6. Some of the supplementary materials are without English-translated versions.
7. As the risk prediction model is intended to facilitate clinical processing, I would suggest including advice for clinicians based on your results in the discussion.

Experimental design

no comment

Validity of the findings

no comment

Reviewer 2 ·

Basic reporting

1. It is necessary to strengthen the connection between sentences and enhance logical coherence within paragraphs.
2. Authors must distinguish between the research in the paper and the author's previous research statements, to reduce confusion (Line 219: Our Study...).
3. The third paragraph in the discussion part contains too much content, so it should be discussed in appropriate paragraphs.
4. Grammar mistakes need to be corrected.
5..In the introduction part, the introduction of the nomogram (Lines 72 -75) would better belong to the method part.
6. The introduction part did not elaborate thoroughly on the background or why this study was conducted.
7. The conclusion part could be more precisely focused on the predictive model built in this study.

Experimental design

1. Please provide detailed inclusion criteria such as diagnostic criteria of CHD as well as references.
2. Line 84: Which retrospective cohort study was referred to? Or did the authors mean that this study was a retrospective cohort study? If the latter, why did in the results section(Line 103-104) authors divide the participants into two groups based on ADR? Besides, there was no further analysis based on the two groups.
3. Provide more details on the statistical methods. The nomogram building process was blurry.
4. How the development and the validation groups were decided.
5. Table 1: education levels would be better arranged in order. Some variables need clear definitions: dyslipidemia, smoking habits, alcohol consumption, basic disease types, combined medication usage, and high-dose drug utilization.
6. This study did not include any laboratory indicators such as liver function indicators. Were those indicators unrelated to the occurrence of ADR? It is necessary to fully explain the reasons for not being included.

Validity of the findings

1. The number of participants included should be mentioned in the result section, as well as the number in the groups of ADR and non-ADR. There is no flow diagram of the study.
2. The authors did not report the number and proportion of different ADR types.
3. What is the optimal cut-off value of the total nomogram scores, and the sensitivity, specificity, positive predictive value of positivity and negative at the optimal cutoff?
3. What are the possible reasons for the lower incidence of ADR in this study compared to literature reports?
4. Is the incidence of various types of adverse reactions in this study consistent with the literature? If not, what are the possible reasons?
5. Line 239 -250: The authors should focus on the utilization of the developed nomogram in this study rather than the general advantages and the assessment of the nomogram.

Additional comments

no comments.

---

## Round 0.2 · Minor Revisions

Dear authors,

The study entitled “Establishment and validation of a risk prediction model for adverse drug reactions in patients with coronary heart disease after taking statins” demonstrated interesting findings using an appropriate methodological approach. However, minor revisions must be clarified in the manuscript. Your article has great potential for publication on PeerJ, but the reviewers have requested substantial changes to be made.

Reviewer 1 ·

Basic reporting

All of my concerns have been addressed, and I have no further comments at this time.

Experimental design

All of my concerns have been addressed, and I have no further comments at this time.

Validity of the findings

All of my concerns have been addressed, and I have no further comments at this time.

Reviewer 2 ·

Basic reporting

1、Three studies are cited in Introduction part to reinforce the background of this study. However, it is advisable to to summarize the findings of the three cited studies, rather than a large paragraph of quoting the original text.
2、The pixels in Figure 1 are too low and the image is not clear。

Experimental design

I understand that ADR is the outcome of the study, but it is not appropriate to group the outcome measures. Grouping is usually done at the beginning of a study in order to compare two groups, such as a randomized controlled study or a case-control study. However, ADR is clearly an outcome to be observed in this retrospective cohort study, and then risk factor analysis and prediction model construction are carried out for this outcome. Please use the correct description. And the number of people who have and do not have ADR should be described in the results.

Validity of the findings

no comments

Additional comments

no comments

---

## Round 0.3 · accepted · Accept

Dear Author,

Congratulations! After your diligent work addressing the reviewers' comments, I am pleased to inform you that your manuscript has been accepted for publication in PeerJ. This version is more concise and formal, enhancing clarity and flow.